# Real-World Evidence of Bevacizumab and Panitumumab Drug Resistance and Drug Ineffectiveness from EudraVigilance Database

**DOI:** 10.3390/cancers17040663

**Published:** 2025-02-16

**Authors:** Razvan Constantin Vonica, Claudiu Morgovan, Anca Butuca, Manuela Pumnea, Remus Calin Cipaian, Adina Frum, Carmen Maximiliana Dobrea, Andreea Loredana Vonica-Tincu, Aliteia-Maria Pacnejer, Florina Batar, Vlad Vornicu, Steliana Ghibu, Felicia Gabriela Gligor

**Affiliations:** 1Preclinical Department, Faculty of Medicine, “Lucian Blaga” University of Sibiu, 550169 Sibiu, Romania; razvanconstantin.vonica@ulbsibiu.ro (R.C.V.); manuela-pia.pumnea@ulbsibiu.ro (M.P.); adina.frum@ulbsibiu.ro (A.F.); carmen.dobrea@ulbsibiu.ro (C.M.D.); loredana.vonica@ulbsibiu.ro (A.L.V.-T.); aliteia.pacnejer@ulbsibiu.ro (A.-M.P.); florina.batar@ulbsibiu.ro (F.B.); felicia.gligor@ulbsibiu.ro (F.G.G.); 2Clinical Department, Faculty of Medicine, “Lucian Blaga” University of Sibiu, 550169 Sibiu, Romania; calin.cipaian@ulbsibiu.ro; 3County Clinical Emergency Hospital of Sibiu, 2-4 Corneliu Coposu Str., 550245 Sibiu, Romania; 4Department of Toxicology, Drug Industry, Management and Legislation, Faculty of Pharmacy, “Victor Babeş” University of Medicine and Pharmacy, 2nd Eftimie Murgu Sq., 300041 Timisoara, Romania; 5Department IX Surgery, Discipline of Oncology, Faculty of Medicine, “Victor Babeş” University of Medicine and Pharmacy, 2nd Eftimie Murgu Sq., 300041 Timisoara, Romania; vornicuvlad91@gmail.com; 6Department of Pharmacology, Physiology and Pathophysiology, Faculty of Pharmacy, “Iuliu Haţieganu” University of Medicine and Pharmacy, 400349 Cluj-Napoca, Romania; steliana.ghibu@umfcluj.ro

**Keywords:** colorectal cancer, bevacizumab, panitumumab, resistance, ineffectiveness, pharmacovigilance, biomarkers, personalized therapy

## Abstract

Colorectal cancer is a leading cause of cancer-related deaths worldwide, and resistance to treatment poses a major challenge to patient survival. This study examines the effectiveness of two commonly used therapies—bevacizumab and panitumumab—in patients with advanced colorectal cancer. By analyzing reports of treatment failure and resistance from the European safety database, the study identifies patterns of reduced drug effectiveness. The findings show that a significant proportion of reported adverse reactions are related to resistance, with panitumumab having a higher likelihood of reported resistance than bevacizumab. These results emphasize the need for personalized treatment strategies based on a patient’s molecular profile. Monitoring drug resistance can help improve therapy selection and increase the chances of successful treatment, ultimately enhancing survival rates and quality of life for patients with colorectal cancer.

## 1. Introduction

Colorectal cancer (CRC) is the third most common cancer in the world, with an average 5-year overall survival (OS) rate of approximately 60% [1]. Early-stage CRC can benefit from surgical treatment, and long-term survival is prolonged compared with metastatic CRC, where survival rates are much lower [2]. In approximately 30% of cases, liver metastases are present at the moment of diagnosis, and in 50–70% of patients, liver metastases develop later, which is the major eventual cause of mortality [3]. Understanding the mechanisms of invasion–metastasis and the role of metastasis-initiating cells (MICs) is essential to understanding the progression and spread of CRC [4].

Since conventional chemotherapy has some limitations (e.g., narrow therapeutic index, severe adverse drug reactions, low bioavailability, high risk of drug resistance, etc.) [5,6] different strategies to improve the selectivity and efficacy of treatment or to reduce risks have been developed. These strategies involve immunotherapy (including antibody drug conjugates), polymer-directed enzyme prodrug therapy, and gene-directed enzyme prodrug therapy (GDEPT), etc. [7,8].

For CRC, as well as for other forms of cancer, the RAS oncogene has frequently been reported to undergo mutations. The treatment of choice in metastatic stage colon cancer depending on the RAS mutation is bevacizumab (BEV) for the mutant RAS and panitumumab (PAN) for the wild-type RAS mutation, combined with chemotherapy [9]. BEV represents an antiangiogenic therapy that has become a major and frequently used therapeutic strategy for various types of tumors [10]. Previous research has underlined the importance of the dose and timing of antiangiogenic treatment, aspects which significantly influence its toxicity [11,12]. Vascular endothelial growth factor (VEGF) withdrawal leads to an increase in extracellular matrix (ECM) deposition in malignant tumors [13,14]. This leads to the activation of molecular event cascades at the cellular level that result in tumor progression and, implicitly, distant dissemination of cancer cells [15,16].

Drug resistance is a significant obstacle in cancer treatment, contributing to the majority of deaths from chemotherapy failure [17]. Studies have identified various mechanisms of resistance, classified into two main categories: non-cellular and cellular mechanisms [18]. Non-cellular mechanisms include extracellular factors such as limited vascular access and tumor microenvironment, while cellular mechanisms refer to drug targets, enzymes, and transport systems within cancer cells, subdivided into classical transport-based and non-classical mechanisms [19]. Investigating these mechanisms and developing strategies to counteract them are crucial for the success of cancer chemotherapeutic treatments [20].

The pathophysiological mechanisms of acquired resistance to antiangiogenic therapy are still unknown, partly due to limited information on the impact of antiangiogenic therapy on the tumor microenvironment [21]. There is recent evidence suggesting a link between mechanotransduction and cellular metabolism [22,23]. Another mechanism of resistance is attributed to the activity of alternative proangiogenic pathways, such as fibroblast growth factor (FGF) and angiopoietin-2 (ANGPT2), which can promote the creation of new blood vessels that will vascularize the tumor, resulting in its progression [24]. Also, platelet-derived growth factor (PDGF) signaling can replace the role of (VEGF) in supporting endothelial cell proliferation [25]. Blocking VEGF has been shown to result in extracellular matrix accumulation and changes in the immune cell population in the tumor microenvironment, resulting in increased macrophage infiltration, immune evasion, and tumor progression [26]. Hypoxia induced by VEGF inhibition is a trigger for tumor metabolic reprogramming, thus tumors switch to aerobic glycolysis-based metabolism, known as the “Warburg effect,” to generate energy and support growth. In addition to glycolysis, fatty acid oxidation and glutamine utilization become essential mechanisms of survival [27].

A study by LaFargue et al. has shown that CD5L (precursor CD5-like antigen) is an important mediator of resistance to anti-VEGF therapy [28]. It was concluded that hypoxia due to prolonged VEGF blockage ultimately leads to CD5L overexpression through upregulation of the transcription factor peroxisome proliferator-activated receptor gamma (PPARG) [29]. Analysis of downstream pathways demonstrated prominent activation of the phosphoinositide 3-kinas/serine/threonine kinase (PI3K/AKT) pathway with increased CD5L signaling in tumor endothelial cells. Importantly, blocking CD5L by using a function-blocking antibody or RNA aptamer restored the response to anti-VEGF therapy [30]. These findings suggest that CD5L plays a central role in the mechanism of adaptive resistance to anti-VEGF therapy. The identification of this protein paves the way for the development of innovative clinical strategies, including the use of RNA aptamers or monoclonal antibodies to prevent resistance to BEV [31].

Adaptive resistance to anti-VEGF therapy is a complex mechanism by which tumors achieve survival. As the molecular pathways involved are analyzed, it is essential to identify key components for the development of next-generation therapies [32]. However, there are limitations, such as the lack of single-cell data and the need for further safety testing before clinical application of CD5L-targeted therapies. Overall, CD5L has been identified as a critical protein in the adaptive response to anti-VEGF therapy, suggesting that targeting could benefit patients receiving BEV therapy [33].

Many studies have provided ample evidence suggesting the presence of aberrant signaling by the epidermal growth factor receptor (EGFR) (the ErbB tyrosine kinase receptor (TKR)), which is correlated with cancer progression [34]. The EGFR activates both the RAS-RAF-MEK and AKT-PI3K pathways, leading to cell proliferation and survival [35]. Over time, anti-EGFR therapies have been developed, such as monoclonal antibodies and small molecule tyrosine kinase inhibitors [36]. Monoclonal antibodies approved for treatment in CRC are cetuximab and PAN. Their mechanism of action consists of binding to the EGFR on the cell surface. They prevent ligand binding and receptor dimerization, thus blocking its activation [37]. This process inhibits signal transduction through the aforementioned pathways.

PAN acts by inhibiting the EGFR, a mechanism underlying intracellular signaling pathways involved in cell survival and proliferation. The wild-type genetic profiles of the KRAS and NRAS genes allows the association of PAN to the treatment [23]. Clinical practice has demonstrated that cetuximab or PAN treatment in patients with mCRC has proven to be successful [38,39]. However, cetuximab or PAN treatment as single agents has been effective in up to 10% in some cases [40], implying resistance to this targeted therapy.

Acquired mutations in the KRAS and NRAS genes are the most well-documented mechanisms of acquired resistance to PAN [41]. These mutations have been shown to reactivate the MAPK/ERK pathway, which plays a role in regulating cell proliferation, differentiation, and survival. Liquid biopsies of circulating tumor DNA (ctDNA) have shown that RAS mutations frequently occur early in resistance, even in cases with an initial wild-type profile. The prevalence of acquired mutations ranges from 40% to 60% of patients that receive PAN treatment [42].

Another mechanism of resistance is represented by the amplification of the MET proto-oncogene receptor tyrosine kinase (MET), which constitutes a key mechanism by which tumor cells can influence the blockage of the EGFR. MET activates PI3K/AKT, signal transducers and activators of transcription 3 (STAT 3) pathways, which play a role in cell growth and survival, but also in metastasis [43].

Protein HER2 amplification or mutations result in intracellular activation of the PI3K/AKT and mitogen-activated protein kinase (MAPK) pathways, which have a stimulatory effect on the EGFR gene, thus counteracting the inhibitory effect of PAN. HER2 has the ability to form heterodimers with the EGFR, which maintains active tumor signaling [44].

Epithelial–mesenchymal transformation (EMT) is a phenotypic change in which epithelial cells lose their polarity and cell adhesion, thus favoring mesenchymal mechanisms such as migration and invasion. The growth of tumor stem cells that contribute to tumor aggressiveness and insensitivity to the mechanism of action of PAN is associated with EMT [45].

The tumor microenvironment may play a key role in acquired resistance. Tumor-associated stromal cells (CAFs) secrete growth factors such as the hepatocyte growth factor (HGF), which activates the MET pathway [46].

Another mechanism of resistance refers to the adoption of a neuroendocrine tumor phenotype, characterized by the loss of epithelial markers and the expression of neuroendocrine markers, such as chromogranin A and synaptophysin. These changes are associated with the activation of the Notch and Hedgehog signaling pathways, which promote cell survival and resistance to therapies [47].

Acquired resistance to PAN has important consequences for subsequent treatment choices. Previous research underlines the importance of the dose and timing of antiangiogenic treatment, aspects which significantly influence its toxicity [11,12], and vascular endothelial growth factor (VEGF) withdrawal leads to an increase in extracellular matrix (ECM) deposition in malignant tumors [13,14]. Because of the mechanism of action of BEV and the fact that it produces tumor hypoxia, it leads to the activation of molecular event cascades at a cellular level that result in tumor progression and, implicitly, distant dissemination of cancer cells [15,16] ctDNA, which may guide therapy adaptation [48].

Exclusive blockade of the VEGF signaling pathway by anti-VEGF monotherapy has proven in clinical practice to be ineffective in advanced cases. Primary or de novo resistance to this type of treatment poses a frequent challenge in cancer management, even when drugs are used [49]. One of the main reasons for the occurrence of this type of resistance is the ability of the angiogenic process to activate alternative signaling pathways, thereby bypassing VEGF inhibition. In addition, it has been suggested that blockade of VEGFRs by tyrosine kinase inhibitors (RTKIs) or specific antibodies may contribute to increased tumor invasiveness and the development of metastases [50].

Several biomarkers have been studied as potential indicators of resistance to anti-angiogenic treatment in various diseases. These include VEGF-D, angiopoietin-2 (Ang2), hepatocyte growth factor (HGF), placental growth factor (PlGF), stromal cell-derived factor-1 (SDF-1), microvascular density (MVD), and the interleukins IL-6 and IL-8 [51,52,53,54,55,56,57]. However, attempts to identify the predictive genetic signature of the VEGF-dependent vascular response have not yielded conclusive results [58].

Many additional randomized trials of BEV in metastatic colorectal cancer (mCRC) have confirmed its efficacy in combination with modern chemotherapy regimens. These include fluorouracil/leucovorin or capecitabine/oxaliplatin (XELOX) in first-line therapy (studies AVF0780g and NO16966), as well as the combination of fluorouracil/leucovorin and oxaliplatin (FOLFOX) in second-line therapy (study E3200). The benefits of BEV administration have also been observed in most lines of therapy, as demonstrated in study ML18147 [59,60,61]. The efficacy of a combined strategy of anti-angiogenic therapy and immunotherapy is highlighted/confirmed by the recent approval of the combination of BEV, atezolizumab and chemotherapy. This decision is based on the results of the IMpower150 trial in patients with metastatic non-squamous metastatic NSCLC (NSq), which demonstrated significant advantages in terms of disease progression survival (PFS) and overall survival (OS) [62]. A Phase I clinical trial (NCT02715531) evaluating the efficacy of the combination of BEV and atezolizumab in the treatment of solid tumors in patients with hepatocellular carcinoma (HCC) showed a significant increase in median progression-free survival (PFS), the primary endpoint of the study. Compared with atezolizumab alone, this combination provided superior outcomes (5.6 months versus 3.4 months, HR 0.55, *p* = 0.0108). In addition, therapeutic responses were absent or very limited with single therapy [63].

Since all medicines can produce side effects, but not all patients will develop side effects of the same type or intensity, the main objective of pharmacovigilance (PV) is to increase safety and maximize therapeutic outcomes. During drug development stages, information about the safety of a drug is sometimes insufficient because clinical trials are conducted in a controlled environment, the number of patients is limited, and they have a specific duration [64]. In this regard, the post-marketing surveillance program is the main tool in detecting serious and rare side effects.

PV is an essential tool for identifying adverse drug reactions (ADRs) and optimizing the safety of drug use [65]. It is a fundamental pillar of drug safety strategies, facilitating measures such as withdrawing certain products from the market, updating labeling and imposing prescribing restrictions. Advanced data analysis plays a crucial role in the early detection of signals of adverse reactions, and its integration with modern information technologies can greatly improve the effectiveness of pharmacovigilance [66]. In addition, many countries have adopted specific regulatory policies in this area, which have enabled them to significantly improve patient safety and the management of risks associated with the use of medicines [67]. PV currently faces several unresolved challenges. Of particular importance are the issues of how to ascertain, collect, confirm, and communicate the best evidence to aid clinical choice for each patient. The identification and monitoring of safety signals, regular review of updated safety reports and review of post-authorization studies are key aspects in this process.

EudraVigilance (EV) is coordinated and managed by the European Medicines Agency. Data are regularly analysed by the PRAC (Pharmacovigilance Risk Assessment Committee), which assesses the signals and can recommend regulatory action accordingly.

Based on real world evidence, the present study aims to evaluate the risk of drug resistance and drug ineffectiveness related to the use of BEV and PAN, monoclonal antibodies used as first line therapy in colorectal cancer.

## 2. Materials and Methods

### 2.1. Study Design

In the present study the risks of ineffectiveness and resistance related to BEV and PAN, reported as spontaneous ADRs, have been evaluated. A comparative analysis of data registered in the EV database (https://www.adrreports.eu/, accessed on 3 December 2024) up to 1 December 2024 has been performed [68]. Data were extracted on 4 December 2024. Firstly, the ratio of the number of ADRs reported for each drug to the total number of Individual Case Safety Reports (ICSRs) and the distribution of ADRs related to ineffectiveness and resistance have been compared. In the EV database, ADRs are reported under different preferred terms (PTs) organized by System Organ Class (SOC). There are 27 SOCs according to the Medical Dictionary for Regulatory Activities (MedDRA). Each PT is coded by MedDRA. For both medical conditions (resistance and ineffectiveness), comparison between the distribution of ADRs by PTs used for reporting (Table 1), respectively, the distribution of ADRs by the outcomes related to BEV and PAN have been performed. Patient status is highlighted by different terms: (i) unfavorable outcome (“Fatal”; “Not recovered/Not resolved”); (ii) favorable outcome (“Recovered/Resolved”; “Recovering/Resolving”); (iii) unknown outcome (“Unknown”).

Subsequently, a disproportionality analysis has been conducted in order to evaluate the probability of reporting the resistance and ineffectiveness of both drugs compared to other drugs used in colorectal cancer. Thus, for the comparison, drugs were chosen which are used as systemic therapy (capecitabine, 5-fluorouracil; oxaliplatin; irinotecan; trifluridine/tipiracil), targeted therapy (adagrasib; aflibercept; regorafenib; sotorasib), and immunotherapy (dostarlimab; nivolumab; pembrolizumab).

### 2.2. Criteria and Selection Process

The data acquisition process was carried out by querying an established pharmacovigilance database: EV. Our study analysed the aggregated data collected from ICSRs available on the open access section of EV. The criteria for selection were the names of the suspected drug BEV or PAN, and for the disproportionality analysis: capecitabine, 5-fluorouracil, trifluridine/tipiracil, irinotecan, oxaliplatin, regorafenib, aflibercept, adagrasib, sotorasib, pembrolizumab, nivolumab, and dostarlimab. From these data, ADRs related to drug ineffectiveness and drug resistance were selected, using specific PTs as criteria for selection. The PTs were identified considering the alternatives proposed by MedDRA—accessible on https://bioportal.bioontology.org/ontologies/MEDDRA/?p=classes (accessed on 25 November 2024)—to describe drug ineffectiveness and resistance [69].

### 2.3. Statistics

According to EMA recommendations, statistics methods used in the EV system for performing disproportionality analysis are represented by the evaluation of the reporting odds ratio (ROR) and 95% confidence interval (95% CI) [70,71]. According to EMA rules, these can be calculated between drugs from the same therapeutic areas or used in similar clinical contexts. A signal is considered disproportionated (the probability of reporting is higher) if the total number of reports is a minimum of 5 and the lower limit of the 95% confidence interval (95% CI) is greater than 1 [71].$$ROR=\frac{a\times d}{b\times c}$$
where:*ROR* = reporting odds ratio;*a* = evaluated ADR for the targeted drug;*b* = other ADRs for the targeted drug;*c* = evaluated ADR for the drug used for comparison;*d* = other ADRs for the drug used for comparison.
95% CI = exp (ln (*ROR*) − 1.96 × SE{ln(*ROR*)}) to exp (ln(*ROR*) + 1.96 × SE{ln(*ROR*)})
where:CI = confidence interval;SE = standard error.$$SE\{ln\left(ROR\right)\}=\sqrt{\frac{1}{a}+\frac{1}{b}+\frac{1}{c}+\frac{1}{d}}$$

The ROR and 95% CI have been calculated using MedCalc software (Version 23.1.7). (MedCalc Software Ltd., Ostend, Belgium) on https://www.medcalc.org/calc/odds_ratio.php (accessed on 5 January 2025) [72].

### 2.4. Ethics

The ICSR does not contain any personal information. Also, non-personal or identifiable data were used to perform the present study. Therefore, ethics board approval was not necessary for the present study [73].

## 3. Results

### 3.1. Comparative Analysis of ADRs Related to PAN and BEV

Until 1 December 2024, a total number of 59,693 ICSRs were uploaded for BEV in the EV database and 7178 for PAN. Also, a high number of ADRs (n = 107,085) were reported for BEV (1.79 ADRs/ICSR) and only 15,051 ADRs for PAN (2.10 ADRs/ICSR) (Figure 1).

Regarding resistance and ineffectiveness, 1.41% referred to BEV resistance or ineffectiveness (n = 1505). The proportion of ADRs related to PAN resistance or ineffectiveness was higher (2.30%, n = 346). More than 93% of ADRs were related to ineffectiveness (BEV: n = 1414, PAN: n = 328) and only some cases were related to resistance (BEV: n = 91, PAN: n = 18) (Figure 2).

#### 3.1.1. Drug Resistance

The distribution of ADRs between both PTs related to resistance is represented in Figure 3. The majority of cases have been reported as “Drug resistance” (BEV: n = 82 and PAN: n = 17).

A high difference is observed regarding the unfavorable outcome. Thus, unfavorable outcomes were reported for 10.99% cases related to BEV resistance (n = 10), and only 0.53% cases related to PAN resistance (n = 1). From these, fatal outcomes represent 80% for BEV and 100% for PAN (Figure 4).

#### 3.1.2. Drug Ineffectiveness

Figure 5 represents the distribution of ADRs related to ineffectiveness. The majority of cases reported were with the following PTs: “Drug ineffective” (BEV: n = 768, PAN: n = 60), “Therapy partial responder” (BEV: n = 179, PAN: n = 172), “Therapy non-responder” (BEV: n = 136, PAN: n = 14), and “Therapeutic product effect incomplete” (BEV: n = 112, PAN: n = 52).

Of total cases related to ineffectiveness, unfavorable outcomes were reported for 181 cases in the BEV group (12.80%) and for 16 cases in the PAN group (4.88%). Fatal cases represent 50.83% for BEV (n = 92) and 25.00% for PAN (n = 4) (Figure 6).

### 3.2. Disproportionality Analysis

#### 3.2.1. Drug Resistance

According to Figure 7, a disproportionate signal could be observed for BEV resistance in comparison with nivolumab (ROR: 6.16, 95% CI: 3.62–10.48), pembrolizumab (ROR: 1.5021, 95% CI: 1.1039–2.0441), and oxaliplatin (ROR: 2.145, 95% CI: 1.4965–3.0745).

PAN has a higher probability of reported resistance in comparison to oxaliplatin (ROR: 3.02, 95% CI: 1.74–5.23), nivolumab (ROR: 8.68, 95% CI: 4.42–17.02), pembrolizumab (ROR: 2.11, 95% CI: 1.26–3.54), cetuximab (ROR: 2.56, 95% CI: 1.30–5.01), and aflibercept (ROR: 2.48, 95% CI: 1.17–5.24). No difference could be observed in comparison to BEV (Figure 8).

#### 3.2.2. Drug Ineffectiveness

A higher probability of reported ineffectiveness was observed for BEV in comparison to capecitabine (ROR: 1.27, 95% CI: 1.16–1.39), oxaliplatin (ROR: 1.93, 95% CI: 1.77–2.11), regorafenib (ROR: 1.89, 95% CI: 1.57–2.28), and nivolumab (ROR: 1.58, 95% CI: 1.46–1.72), and a lower probability of reporting BEV ineffectiveness was observed compared to irinotecan (ROR: 0.83, 95% CI: 0.76–0.92), aflibercept (ROR: 0.74, 95% CI: 0.67–0.83), sotorasib (ROR: 0.47, 95% CI: 0.37–0.61), and pembrolizumab (ROR: 0.87, 95% CI: 0.81–0.93). No statistical difference could be observed compared to trifluridine/tipiracil or 5-fluorouracil (Figure 9).

Ineffectiveness was reported for PAN with a higher probability than the majority of comparators except sotorasib (Figure 10). According to the present data, PAN has a higher probability of reported ineffectiveness than BEV (ROR: 1.6649, 95% CI: 1.4746–1.8797).

## 4. Discussion

Pharmacovigilance is defined as “the science and activities related to the detection, evaluation, understanding and prevention of adverse effects or any other potential problems related to drugs”, including medication errors, ineffectiveness, abuse, off-label use, interactions, and assessment of drug-related mortality [74].

It is important to have better methods of early diagnosis and development of strategies in order to increase the effectiveness of therapeutic approaches. The currently available pharmacological treatments offer very moderate benefits to patients due to the high resistance of tumor cells to anticancer drugs [75].

The results of the present analysis highlight the importance of targeted therapy management in metastatic colorectal cancer (mCRC). The proportion of ADRs related to resistance or therapeutic inefficiency was as follows: specifically, 1.41% (n = 1505) of ADRs were associated with BEV treatment, and 2.30% (n = 346) were associated with PAN. Our study identified that 93% of these ADRs were related to therapeutic inefficiency, as follows: in the case of BEV, n = 91, and PAN, n = 18 (Figure 2). These data may explain the subtle but significant differences between the two therapies. The differences between these two studied drugs are explained by the molecular context and by the different mechanisms of action [76]. BEV is a humanized monoclonal antibody that has the role of inhibiting VEGF, thus preventing angiogenesis [77]. Initially it is effective in controlling the disease but, over time, tumors can develop adaptive resistance mechanisms by activating other proangiogenic pathways, placental growth factor (PIGF), or fibroblast growth factor (FGF) [75]. Another aspect of acquired resistance refers to epigenetic modifications and complex interactions in the tumor microenvironment, which subsequently lead to reduced efficacy of the VEGF blockade [78].

PAN is an anti-EGFR monoclonal antibody, recommended in mCRC with wild-type RAS status [79]. The most common mechanisms of resistance to anti-EGFR therapy are mutations in RAS genes, secondary mutations in BRAF, EGFR or HER2 genes, and also tumor heterogeneity [80].

In our study, BEV has a total of 59,693 ICSRs, with a significant difference compared to PAN with 7178 ICSRs, which may indicate a higher prevalence of reporting, probably correlated with wider clinical use. In the case of PAN, the study shows a lower number of ICSRs, but a higher number of ADRs per ICSR (2.10 compared to 1.79 for BEV). The efficacy of BEV is time-limited and colorectal tumors frequently recur [81]. Recent studies have shown that extracellular vesicles (EVs) may stimulate angiogenesis by mechanisms independent of the VEGF pathway [82]. Colorectal cancer-derived exosomes have also been shown to promote new blood vessel formation via heparin-associated VEGF, which is not inhibited by BEV [83]. In addition, our results indicate that angiogenesis induced by CD133-containing microvesicles (MVs) is resistant to BEV. This suggests that CD133-expressing MVs might play a role in the mechanisms of resistance to antiangiogenic therapy and malignant progression of colorectal cancer [84]. Approximately 60% of colorectal cancer patients develop distant metastases within the first five years after diagnosis, which contributes significantly to increased mortality. A recent study has shown an association between circulating levels of CD133-containing EVs and poor prognosis in patients with metastatic colorectal cancer [85]. Given that CD133-containing EVs promote angiogenesis and are resistant to BEV, investigations to assess to what extent the expression of this marker correlates with patient survival rate according to disease stage are ongoing [86].

The data on resistance for each drug (82 cases for BEV and 17 for PAN), may suggest a lack of therapeutic response expected by clinicians. This type of reporting may be influenced by international pharmacovigilance standards [87]. An important finding is the difference observed regarding the proportion of adverse events associated with resistance, such as 10.99% for BEV and 0.53% for PAN. Although a much higher incidence is observed in the case of BEV, we must take into account the fact that this drug is much more often used in other cancers compared to PAN, which has limited use [88].

Monitoring predictive biomarkers and emerging mutations in monoclonal antibody treatments may identify resistance to these drugs in time [89]. Periodic testing of KRAS mutations and VEGF levels may be useful for subsequent clinical decisions and may favor patient survival [90].

The development of combined therapeutic regimens with an antagonistic effect on multiple oncological pathways simultaneously may be considered a promising strategy to prevent or even delay resistance [91].

Drug inefficiency is a major barrier to the successful treatment of patients diagnosed with cancer [92]. It is important to discuss the molecular basis, pharmacokinetic viability, and therapeutic strategies that may influence drug efficacy [93]. The host’s cellular immunity plays an important role in the mechanisms of resistance to anti-VEGF therapy [94]. The presence of CD8+ cytotoxic T lymphocytes is a favorable prognostic biomarker for cancer patients [95]. High levels of regulatory T cell (T-reg) infiltration are associated with reduced overall survival [96]. These findings emphasize the importance of combination therapies that integrate anti-angiogenic drugs with immune checkpoint inhibitors. Ongoing clinical trials are promising for the introduction of a new era of anti-angiogenic therapies [71].

Alternative angiogenic pathways beyond VEGF signaling influence the efficacy of anti-VEGF therapies [72]. As mentioned earlier, the VEGF pathway plays a critical role in promoting angiogenesis during tumor formation. The identification of alternative angiogenic mechanisms is crucial for addressing anti-VEGF resistance in certain cancers [73,74]. Among these alternative pathways, the angiopoietin-Tie signaling axis angiopoietin-2 (Ang2) is particularly prominent [75]. This vascular-specific receptor tyrosine kinase (RTK) pathway is critical for the development, remodeling and regulation of vascular permeability [76]. Angiopoietin-1 (Ang1) serves as an agonist for the Tie2 receptor, facilitating vascular maturation and stability, whereas Ang2 acts as an antagonist [77,78].

The Tie2 receptor, predominantly expressed in endothelial cells in blood and lymphatic vessels, plays a significant role in the pathophysiology of tumor vasculature [79]. In patients with CRC, elevated serum Ang2 levels correlate with poor responses to BEV therapy, emphasizing the role of Ang2 in anti-VEGF resistance mechanisms.

Another alternative to the VEGF pathway is the platelet-derived growth factor (PDGF) pathway [83]. PDGF is vital for cell growth, survival and motility, particularly in mesenchymal cells during tissue repair and development [84]. Mutations that activate the PDGF receptor contribute to the formation of gastrointestinal stromal tumors (GISTs). In the tumor microenvironment, excessive PDGF signaling supports tumor progression by promoting angiogenesis [85].

Placental growth factor (PlGF), part of the VEGF family, binds to VEGFR1 and its co-receptors NRP-1 and NRP-2, but does not interact with VEGFR2 [86]. PlGF influences endothelial cells, macrophages, bone marrow progenitors and tumor cells by activating VEGFR1-mediated pathways, such as PI3K/Akt and p38 MAPK, independently of VEGF signaling [87]. Several observations have reported overriding PlGF in patients treated with anti-VEGF therapy, suggesting that PlGF could be a therapeutic target for tumors resistant to anti-angiogenic therapy [97].

It has been more than 10 years since FDA approval of the first anti-VEGF drug, BEV, but the mechanism of resistance to this drug remains a challenge for the management of patients diagnosed with cancer [98].

The EGFR is recognized as a key regulator of cell proliferation, differentiation and survival [90], making EGFR-targeted therapies crucial in the treatment of malignancies [91]. The introduction of cetuximab and PAN, which specifically target the EGFR, improved survival outcomes for 10–20% of patients with metastatic colorectal cancer (mCRC) [92]. According to the CRYSTAL trial, combining TSC with FOLFIRI as first-line treatment reduces the risk of disease progression by 15% and extends overall survival by 8.2 months in patients with KRAS wild-type mCRC compared to those receiving FO.

However, resistance to PAN and TSC therapy develops in approximately 80% of patients during treatment [94]. The mechanisms underlying resistance to anti-EGFR monoclonal antibodies (mAbs) have been extensively studied [90]. Mutations in genes in downstream EGFR signaling pathways, such as RAS-RAF-MEK and PI3K/AKT/mTOR, are the main contributors to resistance [95,96]. In addition, activation of feedback mechanisms involving receptors such as ERBB2, MET, and insulin-like growth factor receptor 1 receptor (IGF-1R) may undermine the efficacy of EGFR inhibitors [48]. Emerging evidence also shows that intrinsic factors, such as altered metabolism, autophagy [97], cancer stem cells (CSCs) [98], and epithelial–mesenchymal transition (EMT) [99], are associated with poor outcomes despite anti-EGFR mAb treatment.

Advances in gene detection technologies, including ctDNA, liquid biopsies, and exosomal DNA sequencing, now allow more precise molecular subtyping of tumors [60]. By identifying the unique characteristics of tumor subtypes, targeted therapies have improved treatment outcomes and patient care. For patients with driver gene mutations, combination therapies incorporating multiple targeted inhibitors show significant promise in overcoming resistance to anti-EGFR mAb. Clinical trials have demonstrated that combining EGFR-targeted therapies with inhibitors of BRAF, MET, or MEK may improve survival outcomes and offer new hope [99].

Disproportionality analysis is used in pharmacovigilance studies to detect the safety signals related to ADRs in databases [100]. In particular, in the context of oncology, disproportionality analysis is applied to assess the frequency of reports of drug resistance across different treatments [101]. The application of the analysis can guide researchers and healthcare professionals to identify drugs with a predisposition to the development of resistance, for the purpose of therapeutic decisions or even future research directions [101].

The disproportionality analysis revealed significant results for resistance to BEV compared to the following: nivolumab—ROR of 6.16 (95% CI: 3.62–10.48); pembrolizumab—ROR of 1.5021 (95% CI: 1.1039—2.0441); and oxaliplatin (OXA)—ROR of 2.145 (95% CI: 1.4965–3.0745). These results suggest that BEV has a higher probability of resistance compared to the listed drugs, the highest ratio being observed compared to nivolumab.

It can be noted that while the RORs for BEV versus pembrolizumab (1.5021) and oxaliplatin (2.145) are lower than for nivolumab, they still represent a statistically significant increased probability. Following these findings, it should be recommended to extensively monitor patients treated with BEV to identify the time of development of resistance, combine therapies that can attenuate BEV resistance, and develop in-depth studies of molecular mechanisms of BEV resistance. The selection of patients should be performed according to the panel of mutations, focusing on those least susceptible to developing resistance.

The disproportionality analysis regarding resistance to PAN presents different results compared to several other treatments, such as the following: oxaliplatin—ROR of 3.02 (95% CI: 1.74–5.23); nivolumab—ROR of 8.68 (95% CI: 4.42–17.02); pembrolizumab—ROR of 2.11 (95% CI: 1.26–3.54); cetuximab—ROR of 2.56 (95% CI; 1.30–5.01); and aflibercept—ROR of 2.48 (95% CI: 1.17–5.24). No major differences were observed in the reporting of resistance between PAN and BEV.

Following the disproportionality analysis for PAN resistance, high ROR values were observed, especially compared to nivolumab, indicating that PAN resistance is significantly more frequent than for nivolumab. Regarding treatments such as oxaliplatin, pembrolizumab, cetuximab, and aflibercept, resistance reports for PAN show a moderately higher probability of reporting.

The lack of a significant difference in the reporting of resistance between PAN and BEV demonstrates that these two therapies may have similar resistance profiles, despite different mechanisms of action.

Considering the comparative analysis of resistance to BEV vs. PAN, it is found that both BEV and PAN presented higher rates of reporting of resistance compared to other anticancer drugs, such as nivolumab, pembrolizumab, and oxaliplatin. This may indicate that therapies such as BEV and PAN are more prone to the development of resistance compared to other chemotherapy and immunotherapy options. Regarding differences, PAN demonstrated a higher ROR for resistance compared to nivolumab (ROR: 8.68, 95% CI), and to BEV (ROR: 6.16, 95% CI) too. This analysis indicates a higher frequency of reported resistance for PAN than BEV compared to nivolumab.

The results of the study were interpreted and discussed taking into account the limitations derived from the raw data [102]. There are several pharmacovigilance databases, thus none of them possess all the submitted reports for one active pharmaceutical ingredient. Not all ADRs are reported, underreporting being considered a current phenomenon. The aggregated data provide limited information regarding the exact age or ethnicity of patients. Also, numerical information regarding the prescribed units and the ones administered during the treatment are not available.

The sum of EV reports on adverse reactions to medicines does not represent all available information on their benefits and risks and should therefore not be used alone when making decisions about a patient’s treatment regimen. At the same time, the information on EV reflects the observations and views of the reporter, which refer to suspected events. The information cannot be used to determine the probability of a side effect or to establish a clear cause-and-effect relationship. Pharmacovigilance studies of this kind, based on the analysis of disproportionality of data collected from the spontaneous reporting system do not allow the exact calculation of incidence rates or the determination of the real risk associated with the use of a particular medicine. Such studies do not intend to establish a causal relationship between a drug and a certain adverse reaction, but simply to detect a safety signal [103]. Integrating this type of analysis with other methods such as signal detection algorithms (RORs) can provide a more comprehensive understanding of a medicine’s safety profile and minimize the impact of limitations. On the other hand, the aggregated data accessed for the present study did not allow the construction of a predictive relationship between variables or for calculating the probability of adverse reactions. Thus, an advanced statistical method for evaluating the signals could not be applied.

## 5. Conclusions

The need for closer monitoring of resistance and long-term clinical effects is clear, given the low but significant incidence of resistance to BEV and PAN. Advances in genetics and molecular biology may also provide useful information regarding the mechanisms of resistance to targeted therapies.

The analysis of the information gathered from the EudraVigilance database highlights a relatively small number of treatment resistance reports in BEV and PAN. The distribution of ADRs associated with resistance, as well as the proportion of adverse events, has been observed, revealing important clinical aspects. Given that the mechanisms of drug resistance vary as the disease progresses, the use of personalized and specific combination therapies could benefit from a higher level of resistance control. The resistance to therapies against CRC continues to represent a major challenge in the treatment of this pathology.

In summary, drug resistance in CRC remains a significant barrier to the success of chemotherapy, and the identification of new therapeutic strategies is particularly important. Thus, the study of the fundamental mechanisms that determine drug resistance, as well as the development of resources capable of safely and effectively antagonizing resistance, aimed at blocking these mechanisms, will be essential for patients. Ongoing current research approaches combine anti-angiogenic and immunotherapeutic methods as an alternative for overcoming resistance. Innovative therapeutic strategies will lead to reduced recurrence rates and minimized side effects and will improve the quality of life of patients.

## Figures and Tables

**Figure 1 cancers-17-00663-f001:**
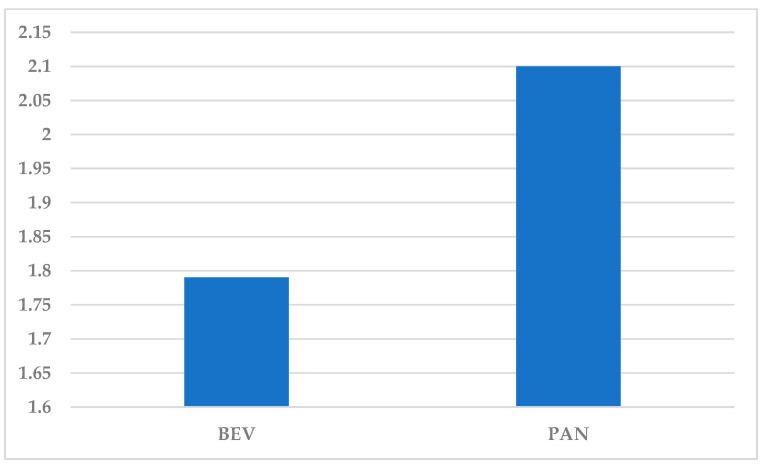
The average of ADRs reported in each ICSR. BEV—bevacizumab; PAN—panitumumab.

**Figure 2 cancers-17-00663-f002:**
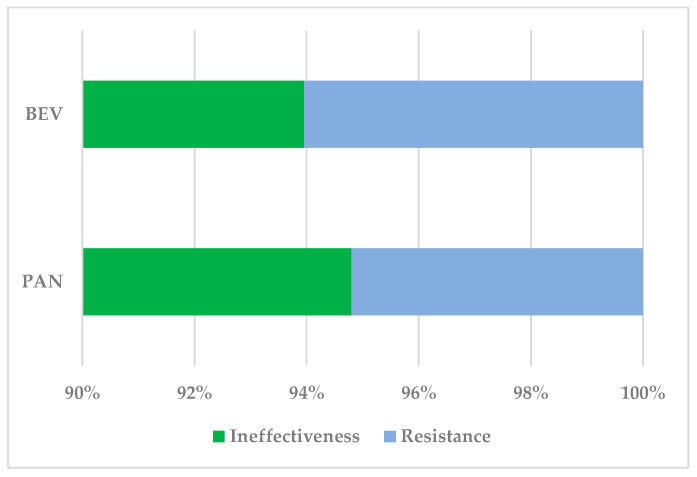
Distribution of ADRs related to ineffectiveness and resistance. BEV—bevacizumab; PAN—panitumumab.

**Figure 3 cancers-17-00663-f003:**
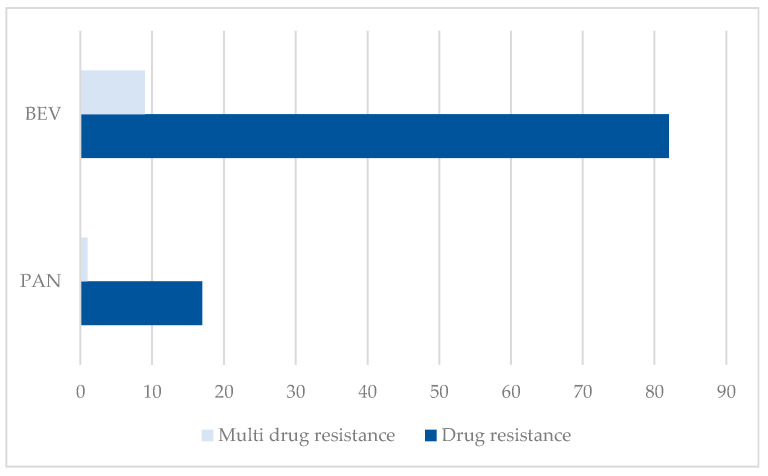
Distribution of ADRs related to resistance by PTs. BEV—bevacizumab; PAN—panitumumab.

**Figure 4 cancers-17-00663-f004:**
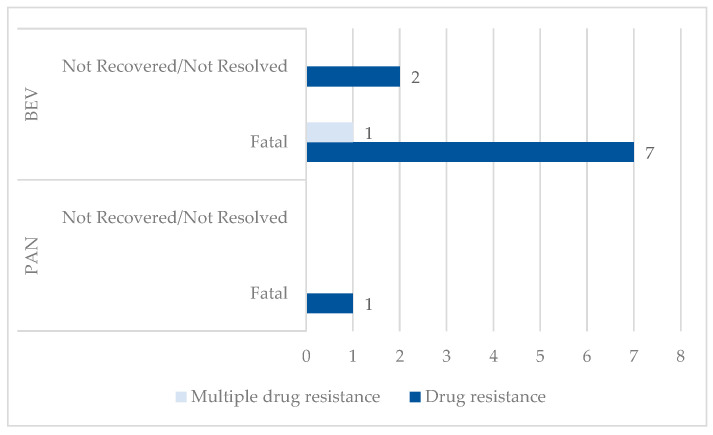
Distribution of ADRs related to resistance by unfavorable outcome. BEV—bevacizumab; PAN—panitumumab.

**Figure 5 cancers-17-00663-f005:**
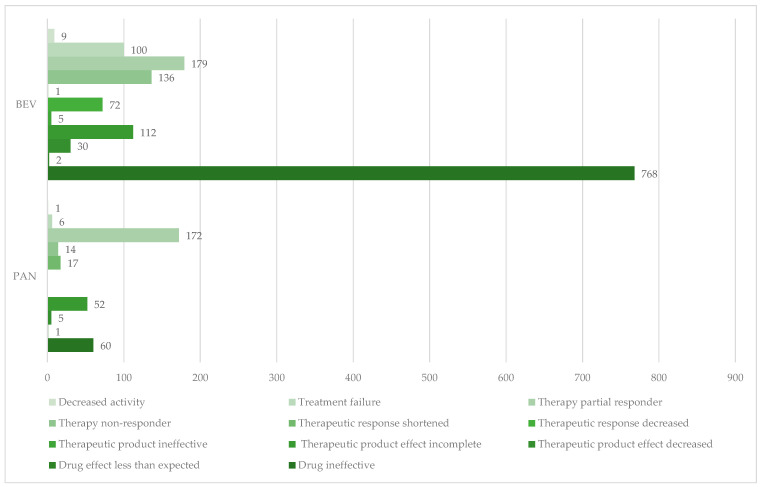
Distribution of ADRs related to ineffectiveness by PTs. BEV—bevacizumab; PAN—panitumumab.

**Figure 6 cancers-17-00663-f006:**
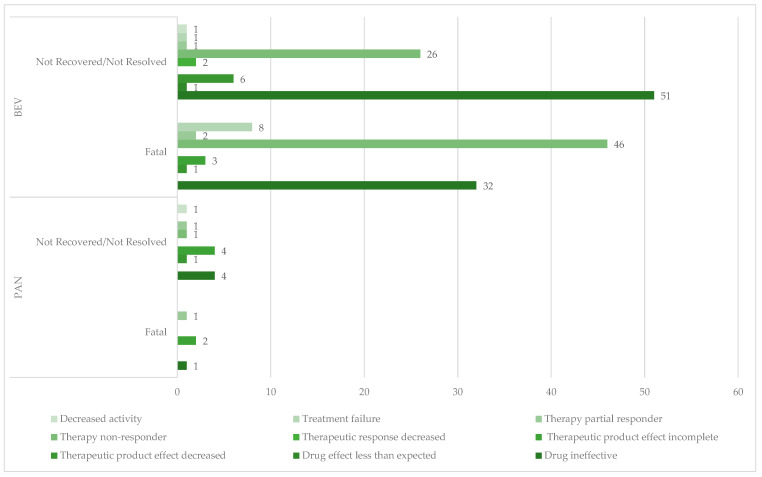
Distribution of ADRs related to ineffectiveness by unfavorable outcome. BEV—bevacizumab; PAN—panitumumab.

**Figure 7 cancers-17-00663-f007:**
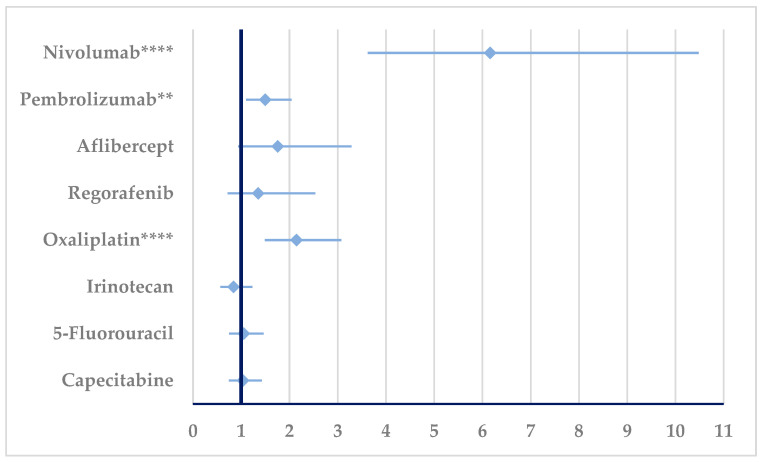
Disproportionality analysis of ADRs related to bevacizumab resistance. ** *p* ≤ 0.01; **** *p* ≤ 0.0001. A higher probability of reporting (lower bound of 95% CI is >1) is observed for bevacizumab resistance compared to nivolumab, pembrolizumab, and oxaliplatin.

**Figure 8 cancers-17-00663-f008:**
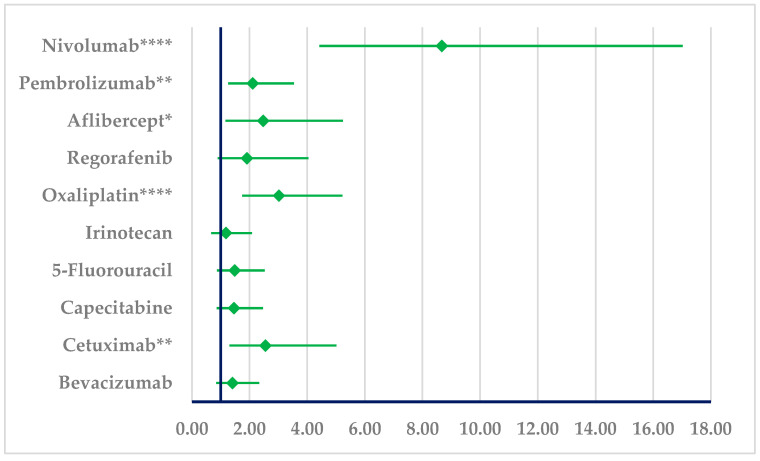
Disproportionality analysis of ADRs related to panitumumab resistance. * *p* ≤ 0.05; ** *p* ≤ 0.01; **** *p* ≤ 0.0001. A higher probability of reporting (lower bound of 95% CI is >1) is observed for panitumumab resistance compared to nivolumab, pembrolizumab, aflibercept, oxaliplatin, and cetuximab.

**Figure 9 cancers-17-00663-f009:**
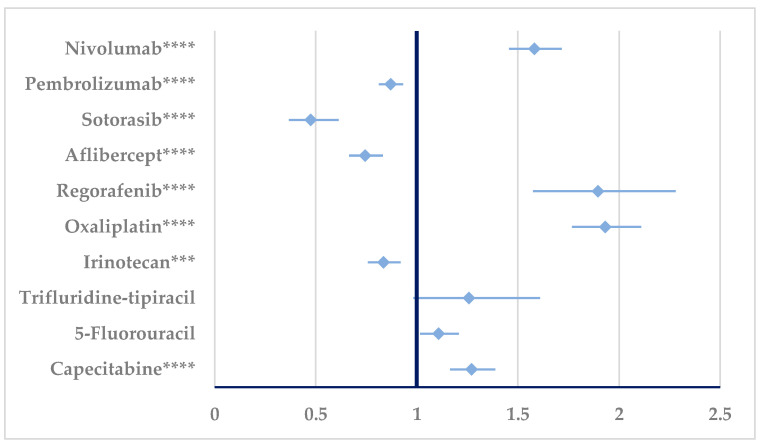
Disproportionality analysis of ADRs related to bevacizumab ineffectiveness. *** *p* ≤ 0.001; **** *p* ≤ 0.0001. A higher probability of reporting (lower bound of 95% CI is >1) is observed for bevacizumab ineffectiveness compared to nivolumab, pembrolizumab, sotorasib, aflibercept, regorafenib, oxaliplatin, irinotecan, and capecitabine.

**Figure 10 cancers-17-00663-f010:**
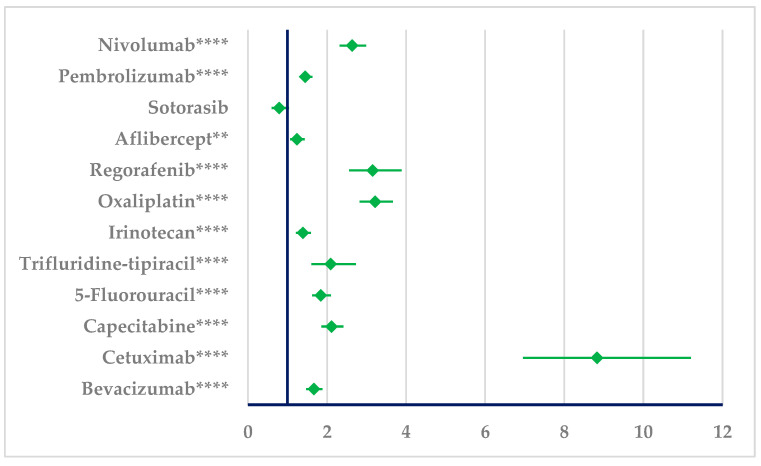
Disproportionality analysis of ADRs related to panitumumab ineffectiveness. ** *p* ≤ 0.01; **** *p* ≤ 0.0001. A higher probability of reporting (lower bound of 95% CI is >1) is observed for panitumumab ineffectiveness compared to nivolumab, pembrolizumab, aflibercept, regorafenib, oxaliplatin, irinotecan, trifluridine/tipiracil, 5-fluorouracil, capecitabine, cetuximab, and bevacizumab.

**Table 1 cancers-17-00663-t001:** PTs used for reporting resistance and ineffectiveness.

Medical Condition	Code	PT
Resistance	10059866	Drug resistance
	10048723	Multiple drug resistance
Ineffectiveness	10013709	Drug ineffective
	10083365	Drug effect less than expected
	10082201	Therapeutic product effect decreased
	10082200	Therapeutic product effect incomplete
	10060769	Therapeutic product ineffective
	10043414	Therapeutic response decreased
	10078575	Therapeutic response shortened
	10051082	Therapy non-responder
	10078115	Therapy partial responder
	10066901	Treatment failure
	10011953	Decreased activity

## Data Availability

All data are contained within the article.

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
