# Peer review of "Real-World Evidence of Bevacizumab and Panitumumab Drug Resistance and Drug Ineffectiveness from EudraVigilance Database"

_cancers, 2025, doi:10.3390/cancers17040663_

Round 1
Reviewer 1 Report
Comments and Suggestions for Authors
-
The study effectively highlights the clinical importance of drug resistance, but further discussion on implications for treatment adaptation could enhance its impact.
-
The methodology section is clear, but specifying the criteria for selecting comparison drugs in the disproportionality analysis would add clarity.
-
The results provide valuable insights; however, a deeper exploration of possible confounding factors in the reported adverse events would strengthen the findings.
-
The discussion addresses key findings well, but integrating more real-world clinical applications of the results would make it more practical for oncologists.
-
The introduction thoroughly presents the background, but briefly summarizing the known mechanisms of drug resistance at the beginning would help set the stage.
-
The statistical analysis is robust, yet adding a brief explanation of why ROR was chosen over other disproportionality measures could be useful for non-specialist readers.
-
The conclusion effectively summarizes the findings, but a stronger emphasis on potential next steps in research or clinical trials would enhance its significance.
-
The figures are helpful in visualizing data, though providing a brief interpretation for each within the text could improve accessibility.
-
While the ethical considerations are addressed, mentioning potential biases in voluntary pharmacovigilance reporting would improve the study’s transparency.
-
The references are extensive and relevant, but citing more recent clinical trials or meta-analyses on drug resistance mechanisms could further validate the findings.
Author Response
- The study effectively highlights the clinical importance of drug resistance, but further discussion on implications for treatment adaptation could enhance its impact.
- The methodology section is clear, but specifying the criteria for selecting comparison drugs in the disproportionality analysis would add clarity.
Thank you. We updated the section with new information.
Lines 188-191: “Thus, for the comparison, were chosen drugs used as systemic therapy (capecitabine, 5-fluorouracil; oxaliplatin; irinotecan; trifluridine-tipiracil), targeted therapy (ada-grasib; aflibercept; regorafenib; sotorasib), and immunotherapy (dostarlimab; nivolu-mab; pembrolizumab), respectivelly.”
- The results provide valuable insights; however, a deeper exploration of possible confounding factors in the reported adverse events would strengthen the findings.
Thank you. The Limitations were extended.
Lines 577-589: “The sum of EV reports on adverse reactions to a medicine does not represent all available information on its benefits and risks and should therefore not be used alone when making decisions about the patient's treatment regimen.
At the same time, the information on this website reflects the observations and views of the reporter, which refer to suspected events. They cannot be used to determine the probability of a side effect or to establish a clear cause-and-effect relationship. Pharmacovigilance studies of this kind, based on the analysis of disproportionality of data collected from the spontaneous reporting system, do not allow the exact calculation of incidence rates or the determination of the real risk associated with the use of a particular medicine. Such studies do not intend to establish a causal relationship between a drug and a certain adverse reaction, but simply to detect a safety signal. (https://pmc.ncbi.nlm.nih.gov/articles/PMC11116264/) Integrating this type of analysis with other methods such as signal detection algorithms (RORs) can provide a more comprehensive understanding of a medicine's safety profile and minimise the impact of limitations.”
- The discussion addresses key findings well, but integrating more real-world clinical applications of the results would make it more practical for oncologists.
Thank you for your observation. The section was improved according to your suggestion.
Lines 190 - 213 : “Several biomarkers have been studied as potential indicators of resistance to anti-angiogenic treatment in various diseases. These include VEGF-D, angiopoietin-2 (Ang2), hepatocyte growth factor (HGF), placental growth factor (PlGF), stromal cell-derived factor-1 (SDF-1), microvascular density (MVD), and the interleukins IL-6 and IL-8 [1–7]. However, attempts to identify the predictive genetic signature of the VEGF-dependent vascular response have not yielded conclusive results [8].
Many additional randomized trials of BEV in metastatic colorectal cancer (mCRC) have confirmed its efficacy in combination with modern chemotherapy regimens. These include fluorouracil/leucovorin or capecitabine/oxaliplatin (XELOX) in first-line therapy (studies AVF0780g and NO16966), as well as the combination with fluorouracil/leucovorin and oxaliplatin (FOLFOX) in second-line therapy (study E3200). The benefits of BEV administration have also been observed in most lines of therapy, as demonstrated in study ML18147 [9–11].
The efficacy of a combined strategy of anti-angiogenic therapy and immunotherapy is highlighted/confirmed by the recent approval of the combination of BEV, atezolizumab and chemotherapy. This decision is based on the results of the IMpower150 trial in patients with metastatic non-squamous metastatic NSCLC (NSq), which demonstrated significant advantages in terms of disease progression survival (PFS) and overall survival (OS) [12].
A Phase I clinical trial (NCT02715531) evaluating the efficacy of the combination of BEV and atezolizumab in the treatment of solid tumors in patients with hepatocellular carcinoma (HCC) showed a significant increase in median progression-free survival (PFS), the primary endpoint of the study. Compared with atezolizumab alone, this combination provided superior outcomes (5.6 months versus 3.4 months, HR 0.55, P = 0.0108). In addition, therapeutic responses were absent or very limited with single therapy [13].
- The introduction thoroughly presents the background, but briefly summarizing the known mechanisms of drug resistance at the beginning would help set the stage.
Thank you. We update information from this section
Lines 182 - 189: “Exclusive blockade of the VEGF signalling pathway by anti-VEGF monotherapy has proven in clinical practice to be ineffective in advanced cases. Primary or de novo resistance to this type of treatment poses a frequent challenge in cancer management, even when drugs are used [1].
One of the main reasons for the occurrence of this type of resistance is the ability of the angiogenic process to activate alternative signalling pathways, thereby bypassing VEGF inhibition. In addition, it has been suggested that blockade of VEGFR by tyrosine kinase inhibitors (RTKIs) or specific antibodies may contribute to increased tumor invasiveness and the development of metastases [2].”
- The statistical analysis is robust, yet adding a brief explanation of why ROR was chosen over other disproportionality measures could be useful for non-specialist readers.
Thank you. We updated the explanation regarding the using of ROR parameter.
Lines 193-195: “According to EMA recommendations, statistics methods used in the EV system for performing the disproportionality analysis are represented by the evaluation of the Reporting Odds Ratio (ROR) and 95% confidence interval (95% CI) [50,51].”
- The conclusion effectively summarizes the findings, but a stronger emphasis on potential next steps in research or clinical trials would enhance its significance.
Thank you for your observation. The section was improved according to your suggestion.
Lines 611 - 614: “Ongoing current research approaches combine anti-angiogenic and immunotherapeutic methods as an alternative to overcome resistance. Innovative therapeutic strategies will lead to reduced recurrence rate, minimized side effects and will improve the quality of life of the patient.”
- The figures are helpful in visualizing data, though providing a brief interpretation for each within the text could improve accessibility.
Thank you. We added the following texts in the caption:
Figure 7: “A higher probability of reporting (lower bound of 95% CI is > 1) is observed for bevacizumab resistance compared to nivolumab, pembrolizumab and oxaliplatin.”
Figure 8: “A higher probability of reporting (lower bound of 95% CI is > 1) is observed for panitumumab resistance compared to nivolumab, pembrolizumab, aflibercept, oxaliplatin, and cetuximab.”
Figure 9: “A higher probability of reporting (lower bound of 95% CI is > 1) is observed for bevacizumab in-effectiveness compared to nivolumab, pembrolizumab, sotorasib, aflibercept, regorafenib, oxaliplatin, irinotecan, capecitabine.”
Figure 10: “A higher probability of reporting (lower bound of 95% CI is > 1) is observed for panitumumab ineffectiveness compared to nivolumab, pembrolizumab, aflibercept, regorafenib, oxaliplatin, irinotecan, trifluridine-tipiracil, 5-fluorouracil, capecitabine, cetuximab, and bevacizuamb.”
- While the ethical considerations are addressed, mentioning potential biases in voluntary pharmacovigilance reporting would improve the study’s transparency.
The sum of EV reports on adverse reactions to a medicine does not represent all available information on its benefits and risks and should therefore not be used alone when making decisions about the patient's treatment regimen.
- The references are extensive and relevant, but citing more recent clinical trials or meta-analyses on drug resistance mechanisms could further validate the findings.
Lines 428 - 442: “The efficacy of BEV is time-limited and colorectal tumors frequently recur [3]. Recent studies have shown that extracellular vesicles (EVs) may stimulate angiogenesis by mechanisms independent of the VEGF pathway [4]. Colorectal cancer-derived exosomes have also been shown to promote new blood vessel formation via heparin-associated VEGF, which is not inhibited by BEV [5]. In addition, our results indicate that angiogenesis induced by CD133-containing microvesicles (MVs) is resistant to BEV. This suggests that CD133-expressing MVs might play a role in the mechanisms of resistance to antiangiogenic therapy and malignant progression of colorectal cancer [6]. Approximately 60% of colorectal cancer patients develop distant metastases within the first five years after diagnosis, which contributes significantly to increased mortality. A recent study has shown an association between circulating levels of CD133-containing EVs and poor prognosis in patients with metastatic colorectal cancer [7]. Given that CD133-containing EVs promote angiogenesis and are resistant to BEV, investigations to assess on what extent the expression of this marker correlates with patient survival rate according to disease stage are ongoing [8].”
Reviewer 2 Report
Comments and Suggestions for Authors
The reviewed article provides a comprehensive analysis of real-world evidence on the resistance and ineffectiveness of bevacizumab (BEV) and panitumumab (PAN) in the treatment of metastatic colorectal cancer (mCRC). Using data from the EudraVigilance (EV) database, the study analyses adverse drug reactions (ADRs) associated with these therapies. Although the introduction provides valuable insights into drug resistance and its mechanisms, it is too detailed. The authors should include a discussion of pharmacovigilance and its importance earlier in the introduction, as this is a key aspect of the study. Emphasizing pharmacovigilance from the outset would better contextualize the importance of ADR analysis and its role in improving patient safety and treatment efficacy. In addition, the study should clearly justify the choice of the EudraVigilance database.
The methodology is well structured and provides a clear comparison between BEV and PAN and other colorectal cancer drugs, providing a broader context for understanding the incidence of ADRs. The study highlights the need for personalized treatment approaches, biomarker monitoring and combination therapy strategies to mitigate resistance - a critical issue in oncology.
An important limitation acknowledged by the authors is the inadequate recording of ADRs in pharmacovigilance databases and the lack of detailed clinical information. The EV database does not contain comprehensive patient demographics, treatment history or genetic profiles, which are critical to understanding individual variations in drug resistance. While the article discusses drug resistance and its mechanisms in detail, it lacks an in-depth discussion of pharmacovigilance.
Overall, this study is a valuable contribution to oncology and pharmacovigilance. However, a restructuring of the introduction to better emphasize pharmacovigilance and an explicit mention of the study’s limitations in the conclusion — particularly the challenges of underreporting and the lack of detailed patient data — would improve the study's coherence and depth.
Comments on the Quality of English LanguageThe English could be improved to more clearly express the research.
Author Response
1. The reviewed article provides a comprehensive analysis of real-world evidence on the resistance and ineffectiveness of bevacizumab (BEV) and panitumumab (PAN) in the treatment of metastatic colorectal cancer (mCRC). Using data from the EudraVigilance (EV) database, the study analyses adverse drug reactions (ADRs) associated with these therapies. Although the introduction provides valuable insights into drug resistance and its mechanisms, it is too detailed. The authors should include a discussion of pharmacovigilance and its importance earlier in the introduction, as this is a key aspect of the study. Emphasizing pharmacovigilance from the outset would better contextualize the importance of ADR analysis and its role in improving patient safety and treatment efficacy. In addition, the study should clearly justify the choice of the EudraVigilance database.
Thank you. We updated the Introduction section by highlighting the importance of pharmacovigilance and the legitimacy of EV as data source.
Lines 172-181: “Since all medicines can produce side effects, but not all patients will develop side effects of the same type or intensity, the main objective of pharmacovigilance is to increase safety and maximise therapeutic outcomes. During drug development stages, information about the safety of a drug is sometimes insufficient because clinical trials are conducted in a controlled environment, the number of patients is limited, and have a specific duration. In this regard, the post-marketing surveillance program is the main tool in detecting serious and rare side effects. EV is coordinated and managed by the European Medicines Agency, Data is regularly analysed by the PRAC (Pharmacovigilance Risk Assessment Committee) which assesses the signals and can recommend regulatory action accordingly.”
2. The methodology is well structured and provides a clear comparison between BEV and PAN and other colorectal cancer drugs, providing a broader context for understanding the incidence of ADRs. The study highlights the need for personalized treatment approaches, biomarker monitoring and combination therapy strategies to mitigate resistance - a critical issue in oncology.
3. An important limitation acknowledged by the authors is the inadequate recording of ADRs in pharmacovigilance databases and the lack of detailed clinical information. The EV database does not contain comprehensive patient demographics, treatment history or genetic profiles, which are critical to understanding individual variations in drug resistance. While the article discusses drug resistance and its mechanisms in detail, it lacks an in-depth discussion of pharmacovigilance.
Thank you. The Limitations were extended.
Lines 577 - 589: “The sum of EV reports on adverse reactions to a medicine does not represent all available information on its benefits and risks and should therefore not be used alone when making decisions about the patient's treatment regimen.
At the same time, the information on this website reflects the observations and views of the rapporteur, which refer to suspected events. They cannot be used to determine the probability of a side effect or to establish a clear cause-and-effect relationship.
Pharmacovigilance studies of this kind, based on the analysis of disproportionality of data collected from the spontaneous reporting system, do not allow the exact calculation of incidence rates or the determination of the real risk associated with the use of a particular medicine. Such studies do not intend to establish a causal relationship between a drug and a certain adverse reaction, but simply to detect a safety signal. (https://pmc.ncbi.nlm.nih.gov/articles/PMC11116264/) Integrating this type of analysis with other methods such as signal detection algorithms (RORs) can provide a more comprehensive understanding of a medicine's safety profile and minimise the impact of limitations.”
4. Overall, this study is a valuable contribution to oncology and pharmacovigilance. However, a restructuring of the introduction to better emphasize pharmacovigilance and an explicit mention of the study’s limitations in the conclusion — particularly the challenges of underreporting and the lack of detailed patient data — would improve the study's coherence and depth.
Lines 221 - 233: “Pharmacovigilance (PV) is an essential tool for identifying adverse drug reactions (ADRs) and optimizing the safety of their use [9]. It is a fundamental pillar of drug safety strategies, facilitating measures such as withdrawing certain products from the market, updating labeling and imposing prescribing restrictions. Advanced data analysis plays a crucial role in the early detection of signals of adverse reactions, and its integration with modern information technologies can greatly improve the effectiveness of pharmacovigilance [10]. In addition, many countries have adopted specific regulatory policies in this area, which have enabled them to significantly improve patient safety and the management of risks associated with the use of medicines [11].
Pharmacovigilance currently faces several unresolved challenges. Of particular importance are the issues of how to ascertain, collect, confirm and communicate the best evidence to aid clinical choice for each patient.The identification and monitoring of safety signals, regular review of updated safety reports and review of post-authorization studies are key aspects in this process.”
Reviewer 3 Report
Comments and Suggestions for Authors
The manuscript describes a study conducted to report on the incidence of drug ineffectiveness and resistance reported in Eudravigilance database with the monoclonal antibodies, Bevacizumab and Panitumumab used in colorecta cancer.
There are some issues which need to be resolved:
Introduction:
1. Page no. 2; Line no. 67-73: The authors explain the mechanism of action of bevacizumab, anti-VEGF antibody; these sentences need reframing to enhance clarity.
Methods:
2. The authors need to provide a detailed account of the data source i.e. Eudravigilance database.
3. What is the coding methodology used in database? Please add a brief description of this.
4. The criteria and process for selection of ICSRs needs to be mentioned in detail. Also, a systematic explanation of the variables to be studied should be added.
5. An important analysis which could have been performed in this study is logistic regression modelling to explore the relationship between patient related characteristics and the reporting of adverse events being studied viz. drug ineffectiveness and resistance. This would further help in identifying such factors contributing to ineffectiveness and suggest strategies in this direction.
6. Did the authors follow any standard guidelines for reporting the study such as READUS-PV Guidelines, The REporting of A Disproportionality analysis for drUg Safety signal detection using ICSRs in PharmacoVigilance.
Results:
7. Information on the year wise distribution of ICSRs included may be added to provide a picture on pattern of reporting such events. Also, the identity of reporter (physician/ pharmacist/ patient etc.) may also be provided.
8. There is duplication of data in text and figures 1, 2 and 3.
9. Some data on characteristics of the study population should also be provided.
Discussion:
10. Other limitations associated with such kind of studies that need to be discussed are misrecognition of reported ADRs and proof of causality.
Comments on the Quality of English Language
Moderate changes in English language required.
Author Response
The manuscript describes a study conducted to report on the incidence of drug ineffectiveness and resistance reported in Eudravigilance database with the monoclonal antibodies, Bevacizumab and Panitumumab used in colorecta cancer.
There are some issues which need to be resolved:
Introduction:
- Page no. 2; Line no. 67-73: The authors explain the mechanism of action of bevacizumab, anti-VEGF antibody; these sentences need reframing to enhance clarity.
Thank you. The paragraph has been rephrased.
Lines 68 - 75: “Previous research underlines the importance of dose and timing of antiangiogenic treatment, aspects which significantly influence its toxicity [11,12]. Vascular endothelial growth factor (VEGF withdrawal leads to an increase in extracellular matrix (ECM) deposition in malignant tumors [13,14]. The tumor hypoxia, it leads to the activation of molecular events cascades at the cellular level that result in tumor progression and, implicitly, distant dissemination of cancer cells [15,16].
Methods:
- The authors need to provide a detailed account of the data source i.e. Eudravigilance database.
Authors thank the reviewer for her/his thoroughness.
Lines 174-176: “. A comparative analysis of data registered in EudraVigilance (EV) database (https://www.adrreports.eu/) , until 1 December 2024, has been performed”
- What is the coding methodology used in database? Please add a brief description of this.
Thank you. We added new information regarding the MedDRA coding. Also, the codes of PTs used in our study have been introduced in Table 1.
Lines 179-181: “In the EV database, ADRs are reported under different preferred terms (PTs) organised by System Organ Class (SOC). There are 27 SOCs according to the Medical Dictionary for Regulatory Activities (MedDRA). Each PT is coded by MedDRA.”
- The criteria and process for selection of ICSRs needs to be mentioned in detail. Also, a systematic explanation of the variables to be studied should be added.
Thank you for pointing this out. We have updated the Methodes section to include your recommendations. Please see lines 195-206.
Lines 195-206: “2.2. Criteria and selection process
The data acquisition process was carried out by querying an established pharmacovigilance database EV. Our study analysed the aggregated data collected from ICSRs available on the open access section of EV. The criteria for selection were the names of the suspected drug BEV or PAN, and for the disproportionality analysis: capecitabine, 5-fluorouracil, trifluridine-tipiracil, irinotecan, oxaliplatin, regorafenib, aflibercept, adagrasib, sotorasib, pembrolizumab, nivolumab, and dostarlimab. From these data, ADRs related to drug ineffectiveness and drug resistance were selected, using specific PTs as criteria for selection. The PTs were identified considering the alternatives proposed by MedDRA, accessible on https://bioportal.bioontology.org/ontologies/MEDDRA/?p=classes to describe drug ineffectiveness and resistance [https://www.mdpi.com/2079-6382/10/12/1512].”
- An important analysis which could have been performed in this study is logistic regression modelling to explore the relationship between patient related characteristics and the reporting of adverse events being studied viz. drug ineffectiveness and resistance. This would further help in identifying such factors contributing to ineffectiveness and suggest strategies in this direction.
Thank you. For the present study, we used aggregated data available on the free version of the EV portal. We could not access the stratified data included in each ICSR. Thus, an advanced statistical method (logistic regression) could not be performed. We introduced this issue as another limitation of the study.
Lines 476-479: “On the other hand, aggregated data accessed for the present study did not allow the construction of a predictive relationship between variables and the probability of an adverse reaction. Thus an advanced statistical method for evaluating the signals could not be applied.“
- Did the authors follow any standard guidelines for reporting the study such as READUS-PV Guidelines, The REporting of A Disproportionality analysis for drUg Safety signal detection using ICSRs in PharmacoVigilance.
In the present study, “EMA/849944/2016 European Medicines Agengy - Science Medicines Health: Screening for adverse reactions in EudraVigilance” guideline was used in order to perform the disproportionality analysis. This guideline is used as reference no.
Results:
- Information on the year wise distribution of ICSRs included may be added to provide a picture on pattern of reporting such events. Also, the identity of reporter (physician/ pharmacist/ patient etc.) may also be provided.
Thank you. The raw data consisted of aggregated reports. The stratification of data by year could not be accessed in the open access version of EV. Our descriptive analysis presented the entire characteristics of the reporter (health-care or non-healthcare professionals). The ICSRs do no distinguish between the different professions of healthcare specialists.
- There is duplication of data in text and figures 1, 2 and 3.
Thank you for pointing this out. We kept the data in the text, and we deleted data labels from Figures 1, 2, 3.
- Some data on characteristics of the study population should also be provided.
The results of the descriptive analysis indicate the entire demographic characteristics that can be accessed from the existing raw data. The limitations of the study have been updated.
Line 491-492: “Aggregated data provides limited information regarding the exact age of the patient or ethnicity.”
Discussion:
- Other limitations associated with such kind of studies that need to be discussed are misrecognition of reported ADRs and proof of causality.
Thank you. The Limitations were extended.
Lines 577 - 589: “The sum of EV reports on adverse reactions to a medicine does not represent all available information on its benefits and risks and should therefore not be used alone when making decisions about the patient's treatment regimen.
At the same time, the information on this website reflects the observations and views of the rapporteur, which refer to suspected events. They cannot be used to determine the probability of a side effect or to establish a clear cause-and-effect relationship. Pharmacovigilance studies of this kind, based on the analysis of disproportionality of data collected from the spontaneous reporting system, do not allow the exact calculation of incidence rates or the determination of the real risk associated with the use of a particular medicine. Such studies do not intend to establish a causal relationship between a drug and a certain adverse reaction, but simply to detect a safety signal. (https://pmc.ncbi.nlm.nih.gov/articles/PMC11116264/) Integrating this type of analysis with other methods such as signal detection algorithms (RORs) can provide a more comprehensive understanding of a medicine's safety profile and minimise the impact of limitations.”
Round 2
Reviewer 1 Report
Comments and Suggestions for Authors
The authors made the required corrections. Manuscript can be accepted.
Comments on the Quality of English LanguageNone
Reviewer 2 Report
Comments and Suggestions for Authors
The authors clearly improved the manuscript, therefore, in my opinion, it is now susceptible for publication
Reviewer 3 Report
Comments and Suggestions for Authors
Changes have been made in the manuscript as desired. No further comments from my side.